# Prevalence of Sarcopenia and Its Association with Quality of Life, Postural Stability, and Past Incidence of Falls in Postmenopausal Women with Osteoporosis: A Cross-Sectional Study

**DOI:** 10.3390/healthcare10020192

**Published:** 2022-01-19

**Authors:** Akira Okayama, Naomi Nakayama, Kaori Kashiwa, Yutaka Horinouchi, Hayato Fukusaki, Hirosuke Nakamura, Satoru Katayama

**Affiliations:** 1Department of General Medicine and Community Health Science, School of Medicine, Hyogo College of Medicine, Sasayama Medical Center, Tanbasasayama 669-2321, Japan; okayama0313@gmail.com (A.O.); kaokashiwa@hyo-med.ac.jp (K.K.); hello.horinouchi@gmail.com (Y.H.); h.fukuzaki0818@gmail.com (H.F.); hcmr.nakamura@gmail.com (H.N.); s-katayama@hyo-med.ac.jp (S.K.); 2Faculty of Health and Nutrition, The University of Shimane, Izumo 693-8550, Japan

**Keywords:** osteoporosis, sarcopenia, AWGS 2019, osteosarcopenia, QOL, postural instability, fragility fracture, malnutrition, GLIM criteria

## Abstract

In this cross-sectional analysis of 61 postmenopausal osteoporosis patients who regularly visited an osteoporosis outpatient clinic, we aimed to clarify the prevalence of sarcopenia and its related clinical factors. Of 61 patients (mean age 77.6 ± 8.1 years), 24 (39.3%) had osteosarcopenia and 37 (60.7%) had osteoporosis alone. Age, nutritional status, and the number of prescribed drugs were associated with the presence of sarcopenia (*p* = 0.002, <0.001, and 0.001, respectively), while bone mineral density (BMD) and % young adult mean BMD were not (*p* = 0.119 and 0.119, respectively). Moreover, patients with osteosarcopenia had lower quality of life (QOL) scores, greater postural instability, and a higher incidence of falls in the past year than patients with osteoporosis alone. In contrast, BMD status showed no correlation with the nutritional status, QOL score, postural instability, or incidence of falls in the past year. In conclusion, the incidence of sarcopenia was relatively high among postmenopausal osteoporosis female patients in an osteoporosis outpatient clinic. Our results suggest that in addition to routine BMD evaluation, assessment and management of sarcopenia may be promoted at osteoporosis outpatient clinics to limit the risk of falls and prevent consequent fragility fractures in osteoporosis patients.

## 1. Introduction

The world’s population is aging rapidly in most regions, and people are living longer. This situation is salient in Japan, whose population became an aging society (wherein older individuals represent ≥7% of the total population) in the 1980s. Since then, the older population has continued to grow, reaching 28.4%, the world’s highest record, in 2020 [1]. Along with this growth, the number of older individuals with multiple comorbidities has increased. Age-related deterioration of the musculoskeletal system is common in older individuals [2,3], and the musculoskeletal system plays an essential role in the activities of daily living (ADL) [4]. Therefore, a disturbed musculoskeletal system may cause falls and fractures [4] and significantly reduce the ability to perform ADL and quality of life (QOL) [5].

Osteoporosis and sarcopenia are two important musculoskeletal conditions adversely affecting the health of older individuals. Osteoporosis is characterized by decreased bone mineral density (BMD) and damage to the microstructure of bone tissue, leading to decreased bone strength and an increased risk of fragility fractures [6]. Fragility fractures are the main adverse consequence of osteoporosis [6], causing disability, an inability to care for oneself, and even death [7,8,9,10]; severe fractures are estimated to cause an additional 30% risk of mortality [9]. An estimated 56 million older individuals worldwide sustained fragility fractures in 2000 [11]. Osteoporosis and consequent fragility fractures impose a heavy economic burden worldwide [12,13,14,15]. In this regard, the prevention of fragility fracture is the main purpose of osteoporosis management. In general, osteoporosis interventions are conducted at an outpatient clinic. Clinical practice guidelines for the pharmacological management of osteoporosis in postmenopausal women have been well formulated and evaluated in Japan and worldwide [16,17]. Antiresorptive (denosumab and bisphosphonates), anabolic (teriparatide and abaloparatide), antisclerostin (romosozumab), and hormonal (hormone replacement therapy and selective estrogen receptor modulators) agents are representative of standardized therapy [17]. Patients with osteoporosis receive standardized medical intervention, and BMD is regularly examined using dual-energy X-ray absorptiometry. BMD reflects the effectiveness of any medical intervention and is a well-known parameter for the risk of fragility fracture in these patients [18].

Conversely, sarcopenia is a skeletal muscle disease characterized by reduced muscle mass and muscle quality as measured by weakened muscle power and physical function [19,20]. Sarcopenia is positively associated with negative events, such as falls, disability, disease, hospitalization, and death [21]. The prevalence of sarcopenia varies depending on the study, from 5% to 17% in community-dwelling older adults [22,23], 14% to 85.4% for those in nursing homes [23,24,25,26], and 10% to 24.3% for those in acute hospitals [27,28]. Sarcopenia is an independent risk factor for adverse outcomes, including difficulties in performing instrumental and basic ADLs [20,29,30], falls [30], osteoporosis [31], longer hospital stays, re-admission [28], and mortality [24]. The International Classification of Diseases, Tenth Revision Clinical Modification (ICD-10-CM) diagnosis code for sarcopenia was issued in 2017. Since then, researchers have focused on the screening, diagnosis, and intervention of older individuals with sarcopenia. The European Workgroup for Sarcopenia first published the diagnostic criteria for sarcopenia in 2010 [32]. The Asian Working Group for Sarcopenia (AWGS) adjusted diagnostic criteria for Asian people in 2014 (Asian Working Group for Sarcopenia criteria, 2014, AWGS 2014) and revised it in 2019 (AWGS 2019) to consider ethnic differences in body composition [33,34]. Compared to osteoporosis, sarcopenia as a disease was established relatively recently. Therefore, no approved pharmacological agents for sarcopenia exist yet. A combination of nutritional and exercise therapy is generally recommended for sarcopenia intervention [19,21].

Interestingly, a meta-analysis has reported that osteoporosis and sarcopenia are closely associated with one another [29]. It has been reported that bones and muscles intensely interact with each other [35]. Osteoporosis and sarcopenia share common risk factors, such as aging, sex, physical inactivity, reduced specific nutrients (i.e., vitamin D), and specific hormones (i.e., growth factors and testosterone) [35]. Osteosarcopenia has recently emerged as a new concept indicating the coexistence of osteoporosis and sarcopenia [36]. Therefore, its epidemiology has not been fully investigated. To develop effective interventions, understanding the relationship between osteosarcopenia and its clinical features and outcomes is important. However, this relationship has not yet been fully investigated.

Therefore, this study was conducted to determine the prevalence of sarcopenia and its related clinical factors among patients with osteoporosis. Moreover, because the main purpose of osteoporosis management is to prevent fragility fractures, the impact of osteosarcopenia on the risk of falls was investigated through objective evaluation of postural instability. This information is likely to facilitate the establishment of more optimal treatment strategies for osteoporosis.

## 2. Materials and Methods

### 2.1. Study Population

This cross-sectional study was conducted at the Hyogo College of Medicine, Sasayama Medical Center, Hyogo, Japan. We enrolled 61 postmenopausal women aged >65 years who had been diagnosed with osteoporosis according to the Japan Osteoporosis Society (JOS) criteria and who regularly visited the osteoporosis outpatient clinic in Sasayama Medical Center for ≥1 continuous year. Patients with other types of osteoporosis (i.e., osteoporosis caused by malignant tumors, steroids, hyperthyroidism, and metabolic diseases) were excluded. We also excluded patients with reduced postural stability due to specific diseases, such as cardiovascular disease and vestibular nerve disorders. Standardized pharmaceutical treatment was provided to all patients based on the osteoporosis guidelines. The study was conducted from 1 April to 30 June 2021, as the frequency of regular visits to outpatient clinics is generally three months for osteoporosis patients. The inclusion criteria were female sex, age ≥65 years, and ability to fill a self-descriptive questionnaire.

The protocol and study design were approved by the Ethics Committee of the Hyogo College of Medicine (IRB: 3899).

### 2.2. Evaluation of the Status of Osteoporosis

Osteoporosis is generally diagnosed according to World Health Organization (WHO) criteria [37] and BMD is measured using dual X ray absorptiometry. According to WHO criteria, osteopenia is defined as follows: a BMD value between −1.0 and −2.5 standard deviation (SD) of that for a young healthy adults or a T-score between −2.5 and −1.0 and osteoporosis is defined by a BMD score of ≤−2.5 SD [37]. JOScriteria, which were used in this study, are widely utilized for diagnosing osteoporosis in clinical settings in Japan. According to the JOS criteria, osteoporosis is defined as a mean BMD below the young adult mean (YAM) of <70% or −2.5 standard deviations [38]. All patients in this study met the JOS criteria at the onset of intervention at the osteoporosis outpatient clinic. Briefly, osteoporosis was diagnosed using BMD criteria or the occurrence of a fragility fracture as follows: (1) BMD ≤ 70% or −2.5 standard deviations of YAM; and (2) the presence of a fragility fracture in either the lumbar spine or proximal femur; or (3) the presence of another fragility fracture and a BMD of <80% of YAM [16]. BMD was measured using dual energy X-ray absorptiometry (DXA, Lunar iDXA, GE Healthcare, Madison, WI, USA).

### 2.3. Diagnosis of Sarcopenia

To diagnose sarcopenia, we applied the AWGS 2019 criteria and used the algorithms for acute to chronic healthcare and clinical research settings [33]. Sarcopenia was assessed during regular medical examinations at an osteoporosis outpatient clinic. Skeletal muscle index (SMI, kg/m^2^) was calculated by dividing limb skeletal muscle mass by the square of the height. We used the latest version of the InBody system for evaluating muscle mass volume (InBody 770; Biospace, Tokyo, Japan). Low muscle mass was defined as a SMI of <5.7 kg/m^2^. Individuals classified as having low muscle mass underwent examination of muscle strength and physical function [34]. Reduced muscle strength was defined as a handgrip strength of <18 kg, in accordance with the cutoff value for women. Handgrip strength was measured using a Smedley hand dynamometer (TTM, Tokyo, Japan) in the non-dominant hand with the patient in a standing or seated position, depending on their ability, and with their arms placed at their sides. The higher value from two consecutive measurements was recorded. Physical function was defined as five sit-to-stand tests of >12 s. Participants were considered to have low physical function when they took longer than 12 s to complete the test or they did not complete it. When decreased muscle strength, decreased physical function, or both were confirmed, the participants were diagnosed with sarcopenia.

### 2.4. Diagnosis of Malnutrition

Malnutrition was screened at the time of the sarcopenia assessment and diagnosed according to the Global Leadership Initiative on Malnutrition (GLIM) criteria [39]. For phenotypic criteria, the cutoff value for low body mass index (BMI) was <18.5 kg/m^2^ for patients aged <70 years or <20 kg/m^2^ for those aged ≧70 years, which are specific references for Asians. Reduced muscle mass was defined by an SMI cutoff value of <5.7 kg/m^2^, incorporated from the AWGS 2019 criteria [33,34]. For the etiologic criteria, we assessed whether participants experienced inflammation from their comorbidities. Congestive heart failure, chronic obstructive pulmonary disease, rheumatoid arthritis, chronic kidney and liver disease, and cancer are representative diseases causing chronic inflammation [40]. Weight loss, as a phenotypic criterion, and reduced food intake or assimilation, as etiologic criteria, were measured through interviews.

### 2.5. Evaluation of the Objective Postural Instability

To objectively assess postural instability, a stabilometric analysis was performed with the following parameters: sway area (cm^2^), sway velocity (cm/s), and total sway length (cm). A triangular force platform (GP-31; ANIMA Co., Tokyo, Japan) was used to objectively assess postural stability using standard test conditions, as previously described [41,42,43]. Each individual stood on a platform in a naturally upright posture with their upper limbs by their sides and feet parallel, with a 2-cm distance between the heel and halluces. This trial was conducted with the eyes open. During measurements, the participants were instructed to gaze at a target 3 m away. The center of pressure (CoP) through the platform was recorded at a sampling frequency of 20 Hz for 30 s using a microcomputer. Dedicated software (GP-31; ANIMA Co., Tokyo, Japan) depicted the stabilograms by plotting the chronological CoP positions and automatically calculating the stabilometric parameters. In this study, the following three parameters were used to evaluate postural stability: (1) sway area (cm^2^), i.e., the area enclosing the circumference of the CoP trajectory (a higher value indicates higher body sway); (2) sway velocity (cm/s), i.e., the mean of the displacement of the CoP trajectory per unit time representing the capacity to stabilize the standing posture (a higher value indicates poorer capacity) [43,44]; and (3) total sway length (cm), i.e., the total length of the locus of the CoP trajectory (a higher value indicates higher body sway) [41,42].

### 2.6. Evaluation of QOL

After examination at the outpatient clinic, patients’ health-related QOL was measured using the EuroQol-5D (EQ-5D) questionnaire, which is a simple generic non-disease-specific questionnaire created by the EuroQoL Group for describing and evaluating health-related QOL [45]. The EQ-5D is a patient-reported outcome measure of how they recognize their current health status. Patients describe their health state in five dimensions (mobility, self-care, usual activities, pain/discomfort, and anxiety/depression). There are three categories of severity in each dimension: no problems, moderate problems, and severe problems. Thus, the resulting health state is defined using a five-digit code. For instance, state 1-2-2-3-3 would indicate no problems in mobility, moderate problems in self-care and usual activities, and severe problems in the dimensions of pain/discomfort and anxiety/depression. The EQ-5D index score was calculated as the sum of five-digit codes, yielding a result within a range of 5–15 [40].

### 2.7. Evaluation of Other Covariants

The prescribed drugs, except for osteoporosis-related drugs, were determined by checking participants’ prescription notebooks.

### 2.8. Statistical Analyses

IBM SPSS Statistics for Windows, version 27 (IBM Corp., Armonk, NY, USA), was used for the data analysis. Continuous variables are reported as mean (SD), and categorical variables are represented as number (%). A sample-estimate univariate analysis was performed to examine the factors associated with sarcopenia and BMD status. The unpaired two-sided t-test or Fisher’s exact test was used to test for differences between the two study groups as appropriate. To determine potential predictors for stabilometric measurements, with adjustment for age as a typical confounder, a multiple linear regression model, including the status of sarcopenia and age, was prepared. The variables were entered into the regression model simultaneously. Statistical significance was set at *p* < 0.05.

## 3. Results

The characteristics of the study participants are summarized in Table 1. A total of 61 women aged ≥65 years (mean age: 77.61 years) were recruited. According to the WHO criteria of BMI, participants whose BMI was <22 kg/m^2^ were categorized as underweight, 22–25 kg/m^2^ as normal weight, and >25 kg/m^2^ as overweight. Most participants (50.8%) were categorized as underweight, while 10 participants (16.4%) were categorized as overweight.

### 3.1. Population and Diagnosis of Sarcopenia and Malnutrition

Of the 61 participants screened using the GLIM criteria, 14 (23.0%) were diagnosed with malnutrition and 24 (39.3%) with sarcopenia, among whom 19 (79%) were categorized as having severe sarcopenia with decreased muscle strength and physical function (Figure 1).

### 3.2. Evaluation of BMD

The mean BMD at lumbar vertebrae 2–4 was 0.95 g/cm^2^, and the mean percentage of young adult BMD was 80.23 at the point of survey. The total period of outpatient follow-up varied depending on the patient. As a prerequisite, all patients were diagnosed with osteoporosis based on the JOS criteria and had already begun receiving standardized pharmaceutical treatments at regular outpatient clinics every three months.

### 3.3. Factors Associated with Sarcopenia

Table 2 shows the factors with a significant difference depend on co-existing sarcopenia. Univariate analysis showed that age was significantly higher in sarcopenia group than non-sarcopenia group (*p* < 0.05) and body composition-related factors, such as BMI and calf circumference (CC) were significantly lower in sarcopenia group than non-sarcopenia group. In addition, the factors related to muscle volume, muscle strength, physical function (such as CC, handgrip strength, and the sit-to-stand tests) and QOL status were significantly lower in sarcopenia group than non-sarcopenia group whereas the prevalence of malnutrition diagnosed by GLIM, the number of non-osteoporosis prescribed drugs, and the number of falls in the past year were significantly higher in sarcopenia group than non-sarcopenia group (*p* < 0.05). The lumbar BMD at lumbar vertebrae 2–4 and percentage of young adult BMD were not significantly different depend on co-existing sarcopenia.

### 3.4. Factors Associated with Reduced BMD

Table 3 shows the factors which showed significant difference depend on BMD status. Patients were divided into two groups according to BMD status: below and above the median. In addition to body composition factors, such as BMI and CC, sarcopenic factors (such as handgrip strength and the sit-to-stand tests) and the prevalence of malnutrition diagnosed by GLIM, the number of non-osteoporosis prescribed drugs, QOL score, and number of falls in 1 year did not show any significant difference depend on BMD status (*p* > 0.05). The BMD at lumbar vertebrae 2–4 and percentage of young adult BMD showed a relationship with BMD status.

### 3.5. Impact of Sarcopenia and BMD Status on the Stabilometric Measurement

Table 4 shows the impact of sarcopenia and BMD status on postural stability. Older individuals with osteosarcopenia showed greater sway velocity and total sway length than those with osteoporosis alone. However, BMD status was not related to postural stability in the univariate analysis.

### 3.6. Multivariate Logistic Regression Analysis of the Effect of Sarcopenia on Postural Stability (Age-Adjusted)

The results of the multivariate linear regression analysis are presented in Table 5. Sarcopenia was identified as an independent factor for postural stability after an adjustment for age (sway velocity: coefficient = 0.264, 95% confidence interval (CI): 0.021–0.506, *p* < 0.05; total sway length: coefficient = 11.078, 95% CI: 4.353–17.802, *p* < 0.05).

## 4. Discussion

This study investigated the prevalence and clinical characteristics of co-existing sarcopenia among postmenopausal older osteoporosis female patients at an osteoporosis outpatient clinic. To the best of our knowledge, this is the first report to clarify that co-existing sarcopenia, rather than the level of BMD, has a significant impact on objectively evaluated postural instability, which is a well-known risk factor for falls and consequent fragility fractures in osteoporosis patients.

Sarcopenia is a syndrome characterized by the loss of skeletal muscle mass and muscle quality and is the main cause of musculoskeletal impairment and loss of functional ability in older individuals. The prevalence of sarcopenia varies depending on the age group, clinical setting, diagnostic tools, and ethnicity. The AWGS criteria are consistent with those of the European Working Group on Sarcopenia in Older People; however, the cutoff values have been adjusted to those of Asian populations according to data from regional cohort studies [33]. We incorporated the AWGS 2019 criteria in this study and discovered that the prevalence of sarcopenia in older women with osteoporosis at an osteoporosis outpatient clinic was higher than that of community-dwelling older individuals and approximately similar to the highest prevalence among the reports from hospitalized patients (39.3%) [27,28,46,47,48,49,50]. This is plausible because osteoporosis and sarcopenia share the same risk factors [35] and are closely associated with each other owing to the significant interaction between bones and muscles [29,35].

We also found that older women with osteosarcopenia had a higher incidence of malnutrition than those with osteoporosis alone. In addition to osteoporosis and sarcopenia, malnutrition is a major cause of adverse health problems in community-dwelling older populations and may result in high mortality, disability, reduced physical function, falls, institutionalization, and hospitalization [51,52,53]. Deprivation of specific nutrients, such as protein and vitamin D, may cause sarcopenia and osteoporosis [35]. The risk of malnutrition increases with age because oral nutritional intake is affected by aging [54,55]. Although the incidence of malnutrition is lower in community-dwelling older individuals, the overall burden is relatively high because older individuals live in their community until an advanced age [56]. The prevalence of malnutrition varies widely depending on the definition applied [57]. The GLIM criteria are the first consensus-based universal definition of malnutrition and can be incorporated in any healthcare setting [57]. The prevalence of malnutrition evaluated by the GLIM criteria and its adverse effects have been reported recently in a community setting [58,59]. Sánchez-Rodríguez et al. reported that 23.4% of community-dwelling older individuals in Belgium were categorized as malnourished according to the GLIM criteria [59]. Moreover, these malnourished older individuals had a 4.4-fold higher mortality risk [58]. In this study, we discovered that 23% of the older women with osteoporosis at the osteoporosis outpatient clinic were malnourished according to the GLIM criteria, a prevalence that is approximately similar to that reported by Sánchez-Rodríguez et al. Interestingly, all malnourished patients had sarcopenia, and malnutrition was significantly associated with sarcopenia, while there was no relationship between malnutrition and BMD. This may indicate that the level of BMD is more dependent on pharmaceutical treatment than on nutritional status once pharmacotherapeutic intervention has started. If attention is paid to BMD alone, this may cause an unnoticeable progression of malnutrition and development of sarcopenia among osteoporosis patients. We should realize that adequate nutrition intake is essential for the management of both osteoporosis and sarcopenia.

Osteoporosis is characterized by a systemic reduction in bone mineralization and a compromised microstructure, leading to bone fragility [60]. The risk of fragility fracture may not solely depend on bone quality and fall risk. Some studies have shown that it is significantly influenced by lower BMD and the risk of falls [61,62]. Sambrook et al. reported that among 2005 individuals in residential care, 82% of the fractures experienced may be attributed to falls. Although fracture rates increased with decreasing bone ultrasound attenuation (BUA) (incidence rate ratio 1.94 for lowest vs. highest BUA tertile, *p* < 0.002), incident falls also affected fracture incidence. Individuals who fell frequently (>3.15 falls per person each year) were 3.35 times more likely to experience a fracture than those who did not fall [62]. In this regard, improving bone density and decreasing the fall risk is the optimal strategy to prevent fragility fractures in patients with osteoporosis.

Notably, greater postural instability is an independent risk factor for osteoporotic fractures [63]. Additionally, a combination of low BMD and high postural sway poses an even higher fracture risk than either factor alone [63]. Human postural control is regulated by the integration of information from visual, vestibular, and proprioceptive input [64,65]. In this study, a stabilometric analysis was conducted with the participant’s eyes open to mimic the natural conditions in their life. Furthermore, patients with a history of pathologies that may have affected vestibular function and vision were excluded from this study. Therefore, the comparison of stabilometric parameters may have reflected differences in postural control capacity due to proprioceptive function. Although it has been recognized that sarcopenia is associated with a high risk of falls, there have been limited studies concerning postural instability among older individuals with sarcopenia. Kim et al. recently reported a significant association between sarcopenia and postural dysfunction among community-dwelling older individuals in Korea [65]. They reported that postural instability is higher in people with sarcopenia, independent of age and sex. This finding is consistent with our result that patients with osteosarcopenia had higher postural instability than those with osteoporosis alone. Here, sway velocity and sway length were significantly greater in patients with osteosarcopenia than in those with osteoporosis alone. Conversely, significant differences in these parameters were not found with respect to BMD status (Table 4). It has been reported that age [66,67], sex [67,68], and BMI [69] affect postural stability in a healthy population. A retrospective study of 1086 cases reported by Simon et. al. showed that lower femoral BMD T-scores, higher age, and male sex were associated with greater sway, as assessed using a stabilometer, and a history of previous fragility fracture showed significantly increased values of sway length compared to that in individuals without a history of fragility fracture [70]. Grip strength was also assessed in their study, but it did not show any relationship with the sway status. They concluded that postural stability is affected by BMD, age, and sex. In this study, the odds ratio for sarcopenia was adjusted for age, and multivariate regression analysis revealed the independent effect of sarcopenia on postural instability in patients with osteoporosis (Table 5). Grip strength assessment is a component in the diagnosis of sarcopenia. The reason their study did not find a relationship between grip strength and sway measurement may be because they assessed grip strength alone, not the status of sarcopenia [70], which requires assessing not only grip strength but also muscle volume. In our study, we assessed sarcopenia using the latest validated, ethnically adjusted criteria, namely AWGS 2019, and the status of sarcopenia was significantly associated with impaired postural stability. Moreover, our study showed that BMD status was not associated with postural stability, even though Simon et al. showed that femoral BMD is independently associated with impaired postural stability. This discrepancy may result from the different backgrounds of the sample populations. Their research population included community-dwelling individuals, while our study included patients diagnosed with and receiving pharmacotherapy for osteoporosis. This discrepancy may reflect that BMD is no longer an indicator of postural instability in patients with osteoporosis receiving pharmaceutical treatment. We did not clarify whether the BMD level is related to the incidence of fragility fractures in this study. However, as most fragility fractures are caused by falling [61], it is important to first focus on reducing the risk of falls. In this study, a prior fall episode in the preceding year was associated with the status of sarcopenia, which was significantly related to postural instability, even after adjusting for age, a well-known risk factor for postural instability [70]. The assessment of sarcopenia is important for distinguishing the population with a higher fall risk, and interventions for sarcopenia should be implemented along with osteoporosis management for these populations.

Our results showed that both postural stability and fall history in the past year were not related to the status of BMD but were related to sarcopenia. This may indicate that treating sarcopenia is important to reduce the risk of falls and prevent fragility fractures in patients with osteoporosis. From this point of view, the evaluation of BMD alone is not sufficient for osteoporosis management at outpatient clinics to prevent falls and consequent fractures. The evaluation of sarcopenia is indispensable and advisable for an optimal strategy of fragility fracture prevention in osteoporosis outpatient clinics.

Considering the above-mentioned circumstances, the recommended management of osteoporosis outpatient clinics should include appropriate lifestyle treatment in addition to pharmacological approaches. Several randomized controlled trials (RCTs) have demonstrated the efficacy of resistance exercise in stimulating osteoblastogenesis and muscle protein synthesis, leading to improvements in the bone microarchitecture, muscle mass and strength, and functional capacity in osteoporotic and sarcopenic seniors [71,72,73]. Selection of appropriate footwear should also be considered as it affects postural and gait stability [74]. Regarding the nutritional approach, it is well established that intake of dietary protein containing abundant levels of leucine, and sufficient vitamin D levels are recommended for muscle and bone health, as well as muscle strength, balance, and functional capacity [75]. Adequate consumption of calcium is also recommended, as it has a role in facilitating muscle contractile force and maintaining bone health [76]. Although specific pharmacological therapy for osteosarcopenia has not yet been developed, it has been reported that the RANK ligand inhibitor denosumab has shown promising effects on muscle and bone [77]. Further RCTs are required to establish future pharmacotherapies for osteosarcopenia.

This study has some limitations. First, this was a retrospective study conducted in a single center, and the sample size was relatively small. Second, we did not compare the prevalence of sarcopenia between subjects with and without osteoporosis. In this regard, our result did not clarify whether the prevalence of sarcopenia in elderly women with osteoporosis was higher than that of without osteoporosis. Third, we analyzed cross-sectional data and the incidence of falls in the past year. Forth, we estimated the risk of falls by assessing postural instability using a stabilometer, which objectively measures the body sway of subjects. Therefore, although we demonstrated that co-existing sarcopenia confers significant impairment in postural stability, future prospective studies are warranted to clarify whether impaired postural stability increases the chance of falls in patients with osteoporosis and whether intervention of sarcopenia improves their postural stability and contributes to a reduction in fall incidence and consequent fragility fractures.

## 5. Conclusions

Our study revealed that sarcopenia co-exists in 39.6% of the postmenopausal older osteoporosis patients treated in an osteoporosis outpatient clinic, and in all such patients with malnutrition. Co-existing sarcopenia was significantly correlated with worse QOL scores and greater postural instability. Moreover, the relationship with postural instability was independent of age, which is a known risk factor for postural instability. Patients with osteoporosis are predisposed to fragility fractures when they fall. Postural instability is a critical risk factor for falls in the older population. Assessment and proper intervention for co-existing sarcopenia can help improve QOL and prevent falls and consequent fragility fractures in postmenopausal patients with osteoporosis.

## Figures and Tables

**Figure 1 healthcare-10-00192-f001:**
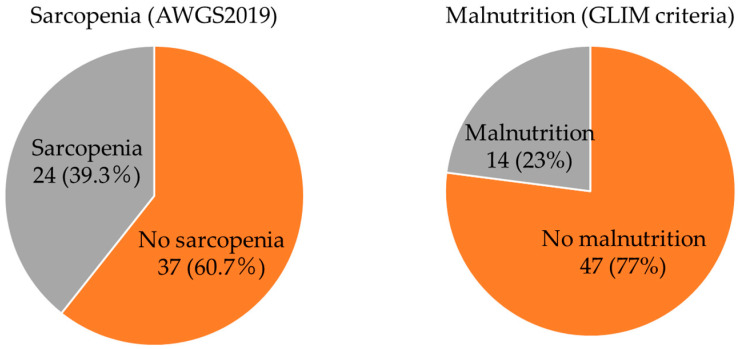
Prevalence of sarcopenia and malnutrition among osteoporosis female older individuals. GLIM, Global Leadership Initiative on Malnutrition; AWGS 2019, Asian Working Group for Sarcopenia 2019.

**Table 1 healthcare-10-00192-t001:** Characteristics of the study population (*n* = 61).

Characteristics	Category	Mean (SD), *n* (%), Median (Range)
Age, years, mean (SD)		77.61 (8.12)
Total muscle mass, kg, mean (SD)		32.31 (3.90)
Body fat rate, %, mean (SD)		21.33 (9.32)
BMI, kg/m^2^, mean (SD)		22.38 (3.14)
BMI category, *n* (%)	Underweight (<22)	31 (50.8)
	Normal weight (22–25)	20 (32.8)
	Overweight (>25)	10 (16.4)
GLIM, *n* (%)	Well nourished	47 (77)
	Malnourished	14 (23)
Sarcopenia, *n* (%)	No sarcopenia	37 (60.7)
	Sarcopenia	24 (39.3)
SMI, kg/m^2^, mean (SD)		5.69 (0.78)
CC, cm, mean (SD)		33.0 (3.24)
Hand grip strength, kg, mean (SD)		18.27 (6.42)
Five times sit-stand test, s, mean (SD)		12.17 (5.11)
Number of prescribed drugs, median (range)		4 (1–11)
Number of falls in one year, times, median (range)		0 (0–2)
Lumbar vertebra 2–4 BMD, g/cm^2^, mean (SD)		0.95 (0.14)
Percentage of young adult BMD, mean (SD)		80.23 (12.16)

SD, standard deviation; BMI, body mass index; GLIM, Global Leadership Initiative on Malnutrition; SMI, skeletal muscle index; CC, calf circumference; BMD, bone mineral density.

**Table 2 healthcare-10-00192-t002:** Comparison of clinical characteristics between patients with and without sarcopenia.

Variable	Non-Sarcopenia (*n* = 37)	Sarcopenia (*n* = 24)	*p*-Value
Age, years, mean (SD)	75.08 (6.71)	81.5 (8.69)	0.002
Percent body fat, %, mean (SD)	21.80 (8.72)	20.62 (10.33)	0.634
BMI, kg/m^2^, mean (SD)	23.34 (3.26)	20.88 (2.30)	0.002
GLIM malnourished, *n* (%)	0 (0)	14 (71)	<0.001
SMI, kg/m^2^, mean (SD)	6.07 (0.69)	5.09 (0.50)	<0.001
CC, cm, mean (SD)	34.43 (3.03)	30.84 (2.20)	<0.001
Hand grip strength, kg, mean (SD)	21.65 (4.79)	13.05 (4.99)	<0.001
Five times sit-stand test, s, mean (SD)	9.325 (2.62)	16.55 (4.93)	<0.001
Number of prescribed drugs			
<4	21	6	0.003
≧4	16	18	
Episode of fall in 1 year			
no	34	13	<0.001
yes	3	11	
EuroQol-5 score, mean (SD)	5.38 (0.54)	7.63 (1.49)	<0.001
Lumbar vertebra 2–4 BMD, g/cm^2^, mean (SD)	0.97 (0.14)	0.92 (0.13)	0.119
Percentage of young adult BMD, mean (SD)	82.19 (12.3)	77.21 (11.5)	0.119

SD, standard deviation; BMI, body mass index; GLIM, Global Leadership Initiative on Malnutrition; CC, calf circumference; EuroQol-5, EuroQol 5-dimension questionnaire; BMD, bone mineral density.

**Table 3 healthcare-10-00192-t003:** Comparison of clinical characteristics between patients with and without BMD reduced below the median.

Variable	Non-Reduced BMD (*n* = 31)	Reduced BMD (*n* = 30)	*p*-Value
Age, years, mean (SD)	77.16 (9.52)	78.07 (6.89)	0.667
Percent body fat, %, mean (SD)	22.91 (9.45)	19.71 (9.06)	0.183
BMI, kg/m^2^, mean (SD)	22.67 (3.12)	22.08 (3.20)	0.468
GLIM malnourished, *n* (%)	6 (19)	8 (26)	0.497
SMI, kg/m^2^, mean (SD)	5.8 (0.80)	5.57 (0.76)	0.27
CC, cm, mean (SD)	33.4 (3.08)	32.63 (3.42)	0.361
Hand grip strength, kg, mean (SD)	18.21 (5.91)	18.33 (7.91)	0.941
Five times sit-stand test, s, mean (SD)	11.74 (4.59)	12.61 (4.15)	0.509
Number of prescribed drugs			
<4	15	12	0.501
≧4	16	18	
Episode of fall in 1 year			
no	23	24	0.579
yes	8	6	
EuroQol- 5 score, mean (SD)	6.32 (1.59)	6.2 (1.54)	0.753
Lumbar vertebra 2–4 BMD, g/cm^2^, mean (SD)	1.06 (0.10)	0.84 (0.08)	0
Percent of young adult BMD, mean (SD)	89.26 (8.67)	70.9 (7.15)	0

BMD, bone mineral density; SD, standard deviation; BMI, body mass index; GLIM, Global Leadership Initiative on Malnutrition; SMI, skeletal muscle index; CC, calf circumference; CC, calf circumference; EuroQol-5, EuroQol 5-dimension questionnaire.

**Table 4 healthcare-10-00192-t004:** Comparison of the stabilometric measurement between the status of sarcopenia and BMD.

**Stabilometric Measurement**	**Non-Sarcopenia (*n* = 37)**	**Sarcopenia (*n* = 24)**	** *p* ** **-Value**
Sway area (cm^2^)	1.59 (0.85)	1.94 (1.31)	0.209
Sway velocity (cm/s)	1.30 (0.31)	1.66 (0.57)	0.003
Total sway length (cm)	38.01 (8.19)	51.58 (6.38)	<0.001
**Stabilometric Measurement**	**Non-Reduced BMD (*n* = 31)**	**Reduced BMD (*n* = 30)**	** *p* ** **-Value**
Sway area (cm^2^)	1.68 (1.15)	1.78 (0.97)	0.7
Sway velocity (cm/s)	1.39 (0.45)	1.5 (0.48)	0.342
Total sway length (cm)	42.34 (12.86)	44.39 (14.66)	0.564

BMD, bone mineral density.

**Table 5 healthcare-10-00192-t005:** Effect of sarcopenia on postural stability (age adjusted).

**Sway Velocity (cm/sec)**			
**Variable**	**Coefficient**	**95% CI**	***p*-Value**
Sarcopenia (vs. non-sarcopenia)	0.264	0.021, 0.506	0.033
Age (per 1 year)	0.015	0.000, 0.029	0.049
**Total sway length (cm)**			
**Variable**	**Coefficient**	**95% CI**	***p*-Value**
Sarcopenia (vs. non-sarcopenia)	11.078	4.353, 17.802	0.002
Age (per 1 year)	0.388	−0.020, 0.795	0.062

95% CI, 95% confidence interval.

## Data Availability

Data that support the findings of this study are available on request from the corresponding author. The data are not publicly available due to privacy or ethical restrictions.

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
