# Peer review of "Prevalence of Sarcopenia and Its Association with Quality of Life, Postural Stability, and Past Incidence of Falls in Postmenopausal Women with Osteoporosis: A Cross-Sectional Study"

_healthcare, 2022, doi:10.3390/healthcare10020192_

Round 1

Reviewer 1 Report

The authors showed that sarcopenia is a significant predictor of postural instability among postmenopausal women with osteoporosis instead of BMD. My concerns are as the following:  Sample size calculation was not indicated. Although the authors have acknowledged this limitation in the discussion, the sample size calculation will convince the readers whether (or not) the study is sufficiently powered. The sentence: "osteopenia is defined as a standard deviation (SD) of BMD scores of-1.0 – - 2.5 for a young, healthy adult (T-score)," This sentence is misleading. It should be rewritten as "osteopenia is defined as a BMD value -1.0 - -2.5 standard deviation (SD) of that for a young healthy adults or a T-score between -2.5 and -1.0".  The statement  "Notably, there is little difference between the JOS and the WHO criteria in T-score values of ≤-2.5 SD" Please elaborate what is the little difference they refer to.  The diagnosis of osteoporosis is different from ISCD guideline that recommended the use of the reference derived from young Caucasian women (https://iscd.org/learn/official-positions/adult-positions/). The impact of the difference in classification on the results should be discussed.  How was the skeletal muscle index calculated? What device was used to measure grip strength? In all tables, put a unit (%) at percent body fat.  In all tables, the data on the number of drugs and number of fractures should be presented as median (range)  Section 3.4, can the authors explain why an arbitrary cut-off of median BMD was used to differentiate subjects with high and low BMD, instead of using classification of osteoporosis? The second paragraph of the discussion should be removed. It is already mentioned in the Introduction. It would be ideal if there was a control group (postmenopausal women without osteoporosis) to illustrate sarcopenia is more prevalent in postmenopausal women with osteoporosis, and it is a better predictor of stability compared to BMD. Can the authors discuss this point?

Author Response

011322

Dear Editor/Reviewers,

Thank you very much for reviewing our manuscript. We found your comments and suggestions to be most valuable and we sincerely appreciate your kindness. Kindly refer to our point-by-point responses to your comments below.

Reviewer1

The authors showed that sarcopenia is a significant predictor of postural instability among postmenopausal women with osteoporosis instead of BMD. My concerns are as the following: 

Comment 1

Sample size calculation was not indicated. Although the authors have acknowledged this limitation in the discussion, the sample size calculation will convince the readers whether (or not) the study is sufficiently powered.

Answer 1: Thank you very much for your comment concerning the sample size calculation. All eligible patients were enrolled in this study. In this regard, we did not calculate sample size in advance of the study. However, we agree that prior determination of the sample size is indeed important for greater reliability. Given that the sample size of this study was relatively small due to its single-centered design, we are planning to undertake a prospective multicentered study with a view to recruiting more patients in future.

Comment 2

The sentence: "osteopenia is defined as a standard deviation (SD) of BMD scores of-1.0 – - 2.5 for a young, healthy adult (T-score)," This sentence is misleading. It should be rewritten as "osteopenia is defined as a BMD value -1.0 - -2.5 standard deviation (SD) of that for a young healthy adults or a T-score between -2.5 and -1.0". 

Answer 2: Thank you very much for your valuable comment. We have modified this sentence.

Comment 3

The statement "Notably, there is little difference between the JOS and the WHO criteria in T-score values of ≤-2.5 SD" Please elaborate what is the little difference they refer to. 

Answer 3: Thank you very much for your comment. JOS criteria are the same as the WHO criteria for BMD. The sentence "Notably, there is little difference between the JOS and the WHO criteria in T-score values of ≤-2.5 SD" has been deleted because it was confusing. In JOS criteria, the risk of fragility-related fractures has been emphasized and criteria have been set for two categories, namely with and without vulnerable fractures. JOS criteria are described in lines 127-134.

Comment 4

The diagnosis of osteoporosis is different from ISCD guideline that recommended the use of the reference derived from young Caucasian women (https://iscd.org/learn/official-positions/adult-positions/). The impact of the difference in classification on the results should be discussed. 

Answer 4: Thank you very much for your comment. It has previously been well established that body composition, including muscle and skeletal bone, varies according to ethnicity. Given this context, we incorporated JOS criteria for the diagnosis of osteoporosis and AWGS 2019 criteria for the diagnosis of sarcopenia. We assumed that it would be appropriate to use corresponding diagnostic tools for a specific ethnic group in this field of research.

Comment 5

How was the skeletal muscle index calculated?

Answer 5: Thank you for your question. We divided limb skeletal muscle mass by the square of the height to calculate the skeletal muscle index (SMI, kg/m2).

The following explanation concerning the skeletal muscle index has been added to the ‘Diagnosis of sarcopenia’ sub-section (lines 140-142):

“Skeletal muscle index (SMI, kg/m2) was calculated by dividing limb skeletal muscle mass by the square of the height.”

Comment 6

What device was used to measure grip strength?

Answer 6: Thank you very much for your question. Handgrip strength was measured using a Smedley hand dynamometer (TTM, Tokyo, Japan) in the non-dominant hand with the patient in a standing or seated position, depending on their ability, and with their arms placed at their sides. The higher value from two consecutive measurements was recorded (lines 147-150).

Comment 7

In all tables, put a unit (%) at percent body fat. 

Answer 7: Thank you very much for your comment. The unit ‘%,’ that is, percentage of body fat, has been included in Tables 1-3, as per your suggestion.

Comment 8

In all tables, the data on the number of drugs and number of fractures should be presented as median (range) 

Answer 8: Thank you for your highlighting this. We are sorry that, in the previous manuscript, Table 1 was incorrect in that it was identical to Table 3. Table 1 has been replaced with the correct version, and median (range) has been included concerning the number of drugs and the number of fractures.

Comment 9

Section 3.4, can the authors explain why an arbitrary cut-off of median BMD was used to differentiate subjects with high and low BMD, instead of using classification of osteoporosis?

Answer 9: Thank you very much for your comment. We used median BMD to classify the patients for analysis. First, all patients had been diagnosed with osteoporosis, based on a reduced BMD, and had been receiving prescription medication in accordance with outpatient clinic guidelines. Clinicians treating these patients regularly examine patient BMD as a marker of prescription effectiveness. Some patients clearly respond well, with a sufficient increase in BMD, whereas others show only a gradual or moderate increase in BMD. As mentioned in our manuscript, falls generally precede fractures. Should a patient fall, those with a reduced BMD are clearly at a higher risk of fracture. In this regard, fall prevention is important in reducing the risk of fracture among patients with osteoporosis in addition to the maintenance of BMD. We used stabilometric analysis to evaluate the risk of falls in these patients. The effectiveness of prescription medication for osteoporosis was not found to be associated with the risk of falling; however, an association was found between sarcopenia status and the risk of falling. We would like to emphasize that clinicians should be vigilant not only to a patient’s BMD status in terms of fracture prevention but also to a patient’s sarcopenia status to further reduce the risk of falls for fracture prevention. For this reason, median BMD was used in this study. I hope our explanation here is a suitable response to your comment.

Comment 10

The second paragraph of the discussion should be removed. It is already mentioned in the Introduction.

Answer 10: Thank you very much for valuable suggestion. The second paragraph has been deleted.

Comment 11

It would be ideal if there was a control group (postmenopausal women without osteoporosis) to illustrate sarcopenia is more prevalent in postmenopausal women with osteoporosis, and it is a better predictor of stability compared to BMD. Can the authors discuss this point?

Answer 11: Thank you very much for your valued comments. You raise a very important point. As you have suggested, it would have been more persuasive to have also included postmenopausal women with no osteoporosis as a control group, to then be able to state: “sarcopenia is prevalent in postmenopausal women with osteoporosis.” We focused on women with osteoporosis because osteoporosis has been recognized more in our recently aged society. The number of women who receive prescription medication for osteoporosis at outpatient clinics has continued to increase as a means of preventing fragility fractures, which carry a significant economic burden globally. Most clinicians consider that BMD follow-up is sufficient as a targeted treatment strategy for osteoporosis to prevent fractures. However, there is first a need to prevent falls and avoid fragility-related fractures. To achieve this, screening for sarcopenia and relevant interventions as may be subsequently required must be considered in osteoporosis outpatient clinics. In this regard, clinicians at osteoporosis outpatient clinics should widen their approach to include assessment and diagnosis of sarcopenia for more effective prevention of fragility fractures in women with osteoporosis who are considered predisposed to fracture. This study involved postmenopausal female patients with osteoporosis for the aforementioned reasons. However, as you suggested, a future study to determine the prevalence of sarcopenia between women with and without osteoporosis would be likely to provide more details concerning the epidemiology of osteosarcopenia. We intend to undertake further research in this regard.

Reviewer 2 Report

BRIEF SUMMARY

This was a cross-sectional study, to investigate the associations between sarcopenia and several clinical factors in women with osteoporosis. Authors report that age, nutritional status, and the number of prescribed drugs were associated with the presence of sarcopenia. Moreover, patients with osteosarcopenia had lower quality of life (QOL) scores, greater postural instability, and a higher incidence of falls in the past year than patients with osteoporosis alone.

I congratulate authors on their work. This is a well-written and technically sound paper with informative figures and tables. Overall, I found the topic timely and clinically important. Before this can be published, I suggest authors to consider my points below.

SPECIFIC COMMENTS

TITLE

The “impact” suggests causation. Since this was not an RCT, please chnage to associations.

ABSTRACT

Line 27: “should be promoted”” Please tone down this statement - data presented do not fully justify such conclusions, as this was a cross sectional study on a small sample size.

INTRODUCTION

In the current form, it is quite difficult to figure out from the information flow in the introduction, why it is important to study this, who will benefit from it, and what is the added value of this paper to current knowledge since there are already similar studies published on the topic. Please clarify.

METHODS

Please provide information about the reliability and validity of all instruments used for the assessment of the outcomes, and if such is not present acknowledge it in a relevant section of the paper.

Please provide definitions for each of the CoP metrics. Also, please state what was considered as stable vs unstable

How did you address statistical bias due to the issue of multiple testing?

Have you checked the normality of data and used appropriate tests depending on it?

Please clarify whether subjects performed practice trials prior to the assessment..

Please provide justification for why the sample size calculation was not performed.

RESULTS

This is a very well-written and structured section.

DISCUSSION

Line 373: “However, as most fragility fractures are caused by falling [65], it is important to first focus on reducing the risk of falls”. I think the reader could benefit from knowing interventions and/or lifestyle modifications that can reduce the risk of falls in older people. I think authors should elaborate on this more. For example, exercise ( https://pubmed.ncbi.nlm.nih.gov/15528779/ ) is shown to reduce fall rates. Moreover, footwear type plays a role in postural and gait stability in older fallers (https://pubmed.ncbi.nlm.nih.gov/33303964/ ) , and therefore should also be considered. Please acknowledge these, and other relevant interventions.

Please provide information on how your results will impact research and/or clinical practice.

Please discuss the generalizability of the results to the wider population with osteoporosis.

Author Response

011322

Dear Editor/Reviewers,

Thank you very much for reviewing our manuscript. We found your comments and suggestions to be most valuable and we sincerely appreciate your kindness. Kindly refer to our point-by-point responses to your comments below.

Reviewer2

This was a cross-sectional study, to investigate the associations between sarcopenia and several clinical factors in women with osteoporosis. Authors report that age, nutritional status, and the number of prescribed drugs were associated with the presence of sarcopenia. Moreover, patients with osteosarcopenia had lower quality of life (QOL) scores, greater postural instability, and a higher incidence of falls in the past year than patients with osteoporosis alone.

I congratulate authors on their work. This is a well-written and technically sound paper with informative figures and tables. Overall, I found the topic timely and clinically important. Before this can be published, I suggest authors to consider my points below.

SPECIFIC COMMENTS

Comment 1

TITLE

The “impact” suggests causation. Since this was not an RCT, please change to associations.

Answer 1: Thank you very much for this suggestion. The title has been changed accordingly.

Comment 2

ABSTRACT

Line 27: “should be promoted”” Please tone down this statement - data presented do not fully justify such conclusions, as this was a cross sectional study on a small sample size.

Answer 2: Thank you very much for your comment. This sentence has been rewritten according to your suggestion.

Comment 3

INTRODUCTION

In the current form, it is quite difficult to figure out from the information flow in the introduction, why it is important to study this, who will benefit from it, and what is the added value of this paper to current knowledge since there are already similar studies published on the topic. Please clarify.

Answer 3: Thank you very much for your comment. Please refer to lines 29-30 and 99-100. Additional text has been added to the Abstract and Introduction to explain how this study adds value to current knowledge for osteoporosis intervention. As mentioned, prevention of falls is essential for reducing the incidence of fragility fractures among patients with osteoporosis because falls tend to precede fragility fractures. However, the risk of falls is not objectively evaluated at osteoporosis outpatient clinics in general. In this regard, we incorporated stabilometric analysis to objectively evaluate the risk of falls in addition to BMD status, which is mandatory for patients with osteoporosis. As such, we consider that the results of our study are likely to better inform clinicians and add value to current osteoporosis interventions and highlight the necessity of sarcopenia interventions in reducing the risk of sustaining fragility fractures.

Comment 4

METHODS

Please provide information about the reliability and validity of all instruments used for the assessment of the outcomes, and if such is not present acknowledge it in a relevant section of the paper.

Answer 4: Thank you very much for your comment. Information concerning all instruments used has now been provided, including DXA, BIA, handgrip dynamometer, and force platform, and have discussed these instruments in the appropriate respective sections. Please refer to lines 135-136, and to lines 168-175.

Comment 5

Please provide definitions for each of the CoP metrics. Also, please state what was considered as stable vs unstable

Answer 5: Thank you for your comment. As we described in the Methods section, the definition of each parameter was, as follows: (i) sway area (cm2), i.e., the area enclosing the circumference of the CoP trajectory (a higher value indicates higher body sway); (ii) sway velocity (cm/s), i.e., the mean of the displacement of the CoP trajectory per unit of time, representing the capacity to stabilize the standing posture (a higher value indicates poorer capacity), and (iii) total sway length (cm), i.e., the total length of the locus of the CoP trajectory (a higher value indicates higher body sway). A cut-off value for the definition of postural instability has yet to be determined. Generally, age affects postural stability. Some studies cited in the Methods section have reported that young and older adult patients show significant differences in terms of average results for these parameters. In this regard, we used age as a confounding factor to analyze the effect of sarcopenia on postural instability.

Comment 6

How did you address statistical bias due to the issue of multiple testing?

Have you checked the normality of data and used appropriate tests depending on it?

Answer 6: Thank you very much for your questions. We changed our statistical methods concerning the univariate analysis of the number of drugs and the number of falls in the past year. Tables 2 and 3 have now been replaced with new tables. The results were the same as our previous results; therefore, we assumed that they did not affect our interpretations.

Comment 7

Please clarify whether subjects performed practice trials prior to the assessment.

Answer 7: Thank you for your comment. All study patients had been diagnosed with primary osteoporosis using appropriate diagnostic tools prior to commencing pharmacologic treatment at the outpatient clinic. They received regular BMD examinations along with examinations required for a diagnosis of sarcopenia at the follow-up outpatient clinic visit.

Comment8

Please provide justification for why the sample size calculation was not performed.

Answer 8: Thank you very much for your comment. We enrolled all eligible patients in this study. In this regard, we did not calculate sample size in advance of the study. However, prior determination of the sample size is indeed important for greater reliability. As the sample size of this study was relatively small due to its single-centered design, we are planning to undertake a prospective multicentered study in future to recruit more patients.

Comment 9

RESULTS

This is a very well-written and structured section.

Answer 9: Thank you very much for your comment.

Comment 10

DISCUSSION

Line 373: “However, as most fragility fractures are caused by falling [65], it is important to first focus on reducing the risk of falls”. I think the reader could benefit from knowing interventions and/or lifestyle modifications that can reduce the risk of falls in older people. I think authors should elaborate on this more. For example, exercise ( https://pubmed.ncbi.nlm.nih.gov/15528779/ ) is shown to reduce fall rates. Moreover, footwear type plays a role in postural and gait stability in older fallers (https://pubmed.ncbi.nlm.nih.gov/33303964/ ) , and therefore should also be considered. Please acknowledge these, and other relevant interventions.

Answer 10: Thank you very much for your suggestion. It is indeed important knowledge for preventing falls in real life. The second article that you suggested has now been included in the discussion and the effectiveness of resistance exercise has also been noted.

Comment 11

Please provide information on how your results will impact research and/or clinical practice.

Answer 11: Thank you very much for your comment. Currently, in the osteoporosis outpatient clinic, BMD is routinely evaluated because it is an indicator of the effectiveness of osteoporosis pharmacotherapy. However, the evaluation of sarcopenic status is not widely implemented there because its necessity has not fully been recognized. Our study findings showed that, in addition to increased BMD, intervention for sarcopenia is important in terms of reducing the risk of falls to prevent fragility fractures among patients with osteoporosis. This finding may prompt clinicians to also assess the sarcopenia status of their patients at risk of falls in their local settings to prevent falls and subsequent frailty-related fractures.

Comment 12

Please discuss the generalizability of the results to the wider population with osteoporosis.

Answer 12: Thank you very much for your comment. As we mentioned in the Discussion section, future prospective studies are warranted to determine whether impaired postural stability increases the risk of falls in patients with osteoporosis and whether interventions for sarcopenia improve their postural stability and contribute to a reduction in fall incidence and consequent fragility fractures among women with osteoporosis. We also mentioned in the Introduction section that an estimated 56 million older adults worldwide sustained fragility fractures in 2000 and that this imposes a heavy economic burden worldwide. We consider that targeted intervention for osteosarcopenia has the potential to contribute to improved geriatric health and lower medical costs.

Round 2

Reviewer 1 Report

Thank you for addressing my previous comments.

Abstract/Introduction: when stating the aim, I recommend "we aimed to/this study was conducted to determine" instead of "clarify"

Since the authors are using univariate analysis for results in Table 2 and 3, I don't recommend the description of "there is an association between X and Y" or "X is associated with Y". The more accurate interpretation is "X is significantly higher/lower in A than B group". I hope you will consider this and reword the results section.

Please add the coefficient of variations of the DXA machine.

On the comment of the lack of non-osteoporotic control subjects, I hope the authors can include it as part of the limitations.

Thank you.

Author Response

011522

Dear Editor/Reviewers,

Thank you very much for reviewing our manuscript again. We found your comments and suggestions to be most valuable and we sincerely appreciate your kindness. Kindly refer to our point-by-point responses to your comments below.

Reviewer1

Comment 1

Abstract/Introduction: when stating the aim, I recommend "we aimed to/this study was conducted to determine" instead of "clarify"

Answer 1:

Thank you very much for your valuable comment. We have modified sentence according to your suggestion.

Comment 2

Since the authors are using univariate analysis for results in Table 2 and 3, I don't recommend the description of "there is an association between X and Y" or "X is associated with Y". The more accurate interpretation is "X is significantly higher/lower in A than B group". I hope you will consider this and reword the results section.

Answer 2:

Thank you very much for your valuable comment. We have reworded sentences according to your suggestion. Please refer line 242-254, 260, and 265.

Comment 3

Please add the coefficient of variations of the DXA machine.

Answer 3:

Thank you very much for your comment. The coefficient of variations of BMD is 0.14.

Comment 4

On the comment of the lack of non-osteoporotic control subjects, I hope the authors can include it as part of the limitations.

Answer 4:

Thank you very much for your comment. I added sentences according to your suggestion in line 428-431.
